# Crossroads between Autoimmunity and COVID-19 in Lung Transplant Recipients

**DOI:** 10.3390/v15102045

**Published:** 2023-10-03

**Authors:** Madhusudhanan Narasimhan, Alagarraju Muthukumar, Kavithalakshmi Sataranatarajan, Lenin Mahimainathan, Luke Mahan, Irina Timofte, Srinivas Bollineni, John Joerns, Song Zhang, April Gorman, Amit Banga, Manish Mohanka, Fernando Torres, Adrian Lawrence, Mohanakumar Thalachallour, Vaidehi Kaza

**Affiliations:** 1Department of Pathology, University of Texas Southwestern Medical Center, Dallas, TX 75390, USA; madhusudhanan.narasimhan@utsouthwestern.edu (M.N.); alagarraju.muthukumar@utsouthwestern.edu (A.M.); kavithalakshmi.sataranatarajan@utsouthwestern.edu (K.S.); lenin.mahimainathan@utsouthwestern.edu (L.M.); 2Department of Internal Medicine, University of Texas Southwestern Medical Center, Dallas, TX 75390, USA; luke.mahan@utsouthwestern.edu (L.M.); irina.timofte@utsouthwestern.edu (I.T.); srinivas.bollineni@utsouthwestern.edu (S.B.); john.joerns@utsouthwestern.edu (J.J.); amit.banga@utsouthwestern.edu (A.B.); manish.mohanka@utsouthwestern.edu (M.M.); fernando.torres@utsouthwestern.edu (F.T.); adrian.lawrence@utsouthwestern.edu (A.L.); 3Peter O’Donnell Jr. School of Public Health, University of Texas Southwestern Medical Center, Dallas, TX 75390, USA; song.zhang@utsouthwestern.edu (S.Z.); april.gorman@utsouthwestern.edu (A.G.); 4Norton Thoracic Institute, St. Joseph’s Hospital and Medical Center, Phoenix, AZ 85013, USA; tm.kumar@dignityhealth.org

**Keywords:** auto-antibodies, lung transplantation, chronic allograft dysfunction, acute rejection

## Abstract

The presence of a certain group of auto-antibodies (AAbs) is known to correlate with the severity of COVID-19. It is, however, unknown if such AAbs are prevalent and impact COVID-19-related outcomes in lung transplant recipients (LTRs) who are immunosuppressed. We performed a retrospective study of LTRs with COVID-19 and analyzed samples before and after COVID-19 for IgG AAbs. AAbs analysis was carried out using autoimmune and coronavirus microarray and the resulting cross-sectional differences in Ab-scores and clinical variables were analyzed using Fischer’s Exact test for categorical variables and a paired *t*-test for continuous variables. Linear regression was used to analyze the differences in Ab-scores and COVID-19 severity. LTRs with non-severe [NS gp (*n* = 10)], and severe [S gp (*n* = 8)] COVID-19 disease were included. Ferritin and acute respiratory failure were higher in the S group (*p* = 0.03; *p* < 0.0001). Among the AAbs analyzed, interferon-related AAbs (IFN-alpha2, IFN-beta, IFN lamba, IFN-epsilon), eight interleukin-related AAbs, and several tissue-related AAbs were also found to be changed significantly from pre- to post-COVID-19 (*p* < 0.05). IFN-lambda (*p* = 0.03) and IL-22 (*p* = 0.002) were significantly associated with COVID-19 severity and remained significant in linear regression analysis while controlling for other variables. AAbs are common in LTRs, and certain groups of antibodies are particularly enhanced in LTRs with severe COVID-19. Preliminary observations of this study need to be confirmed by a larger sample size.

## 1. Introduction

Coronavirus disease 2019 (COVID-19), caused by severe acute respiratory syndrome coronavirus-2 (SARS-CoV-2), in lung transplant recipients (LTRs) results in a case fatality rate of 10–46%, as reported in several prior studies [1,2]. However, there is considerable variability in clinical presentation. Some remain mildly symptomatic, while some develop critical illness despite being on a similar immunosuppressive regimen. Some of the key questions include identifying the determinants of this variable clinical spectrum. Several viral and host factors influence the clinical phenotype. Recent evidence suggests that antibodies to self-antigens known as auto-antibodies (AAbs) participate in graft injury patterns after lung transplantation [3,4]. Pre-existing immunity and host genetic factors are now recognized as important determinants of the severity of COVID-19. Several studies have implicated the presence of AAbs against certain groups of cytokines, such as interferons (IFNs) in patients with severe COVID-19 [5]. Several cytokines are involved in innate and adaptive immunity and can interfere with viral replication and modulate host inflammatory response. IFN response constitutes one such example, where it is the major first line of defense against SARS-CoV-2 virus and impaired reaction leads to severe disease. Another recent study reported enhanced IgA response in severe COVID-19 disease [6]. Bastard et al. [5] identified neutralizing AAbs against type I IFN in up to 13.7% of patients with severe COVID-19 disease that were absent in patients with mild disease and were only found in 0.33% (4/1227) of healthy individuals not exposed to SARS-CoV-2. However, it is unknown whether such a phenomenon exists in the setting of immunosuppression in LTRs, and is unclear if the presence of such AAbs contributes to the severity of COVID-19 in LTRs. We sought to evaluate the presence of AAbs in LTRs who were diagnosed with COVID-19 by comparing measurements before and after COVID-19 and correlated AAbs to the severity of COVID-19 disease.

## 2. Materials and Methods

### 2.1. Study Population

After institutional approval, the study was conducted utilizing the samples provided by the UT Southwestern Biorepository Center that were collected for all COVID-19-positive LTRs. Pre-COVID-19 samples were retrieved within three months before a diagnosis of COVID-19 from our histocompatibility lab (HLA). Inclusion criteria are LTRs with COVID-19 with pre-COVID-19 samples available within three months of COVID-19 diagnosis. Exclusion criteria were those who did not consent for sample collection, re-transplantation, dual organ transplant recipients, and if pre-COVID-19 samples were unavailable from the HLA lab. Pre-COVID-19 samples collected as standard clinical care and stored in the HLA lab for patients included in the study were retrieved. Various demographic, clinical, and outcome variables for the cohort were recorded from the electronic medical record. Donor-specific antibodies (DSA), cumulative acute rejection (CAR), defined as cumulative acute rejections since transplant, pre-COVID-19 spirometry done during the last visit before COVID-19 diagnosis, presence of chronic lung allograft dysfunction was recorded systematically. Patients with COVID-19 are considered to have severe illness if they have SpO_2_ < 94% on room air at sea level, PaO_2_/FiO_2_ < 300 mm Hg, a respiratory rate >30 breaths/min, or lung infiltrates > 50%. COVID-19 severity was defined per the Centers for Disease Control guidelines.(https://www.covid19treatmentguidelines.nih.gov/overview/clinical-spectrum/) accessed on 6 March 2023.

### 2.2. Post-Transplant Management Protocol

All recipients were treated with a standardized maintenance immunosuppressive regimen consisting of tacrolimus, azathioprine/Cellcept, and prednisone. DSA was checked monthly in the first year post-transplant, then every three months for the next 2 years, and yearly thereafter (refer to the supplement for the attached protocol). The management of patients with COVID-19 was based on a predefined protocol developed with available evidence, expert guidance, and multidisciplinary input from the lung transplant team [7].

### 2.3. Autoantigen Microarray Analysis

Sera of all subjects were aliquoted and stored at −80 °C. Autoantigen microarrays were manufactured in the microarray core facility of the University of Texas Southwestern Medical Center, Dallas, TX, USA. A selection of 120 autoantigens was made based on published literature, previously known AAbs in various immune-related diseases, cancer, allergic disease, etc. [8]. A total of 120 antigens were ordered from different vendors and autoantigen array chips were manufactured in the microarray core of UT Southwestern using a nanoplotter microarray printer. The antigens were selected based on a review of the literature and were tested for their specificity with corresponding antibodies on our autoantigen array system. Most of the antigens are recombinant proteins expressed in E. coli, insect, or mammalian cell expression systems. Antigens used in microarray are validated with ELISA or Western blot, with good correlation in multiple prior published studies [8,9,10,11]. Two positive controls, each with 4 various dilutions (anti-Ig control 1:2, anti-Ig control 1:4, anti-Ig control 1:8, anti-Ig control 1:16, Ig control 1:2, Ig control 1:4, Ig control 1:8, Ig control 1:16), were also imprinted on the arrays as positive controls. Mouse serum samples were first treated with deoxyribonuclease I (DNAse I) to remove free DNA and then applied onto autoantigen arrays with 1:50 dilution. The AAbs binding to the antigens on the array were detected with cy3-labeled anti-human IgG, and the array slides were scanned with a Genepix 4400A scanner with laser wavelengths 532 nm for cy3 to generate Tiff images. Genepix Pro 7.0 software was used to analyze the image and generate the Genepix report (GPR) files (Molecular Devices, Sunnyvale, CA, USA). The net fluorescent intensity (NFI) of each antigen was generated by subtracting the local background and negative control (Phosphate buffered saline or PBS) signal. A higher NFI indicates a higher signal over background noise. NFI was normalized by a robust linear model using built-in Ig control with various dilutions [11].

### 2.4. Statistical Changes

Database management, validity interrogation, and statistical analysis were performed using the R Project (https://www.R-project.org). Descriptive analysis was performed where the continuous variables were presented as mean and standard deviation and categorical variables by frequency and proportion. A total of 95% confidence intervals are reported for estimated parameters. Comparisons between groups were performed based on the two-sample *t*-test and Fisher’s Exact test. Changes in AAbs before and after COVID-19 are evaluated based on the paired *t*-test. Linear regression models were employed to assess the association between the changes before and after COVID-19 and severity, controlling for time from transplant and history of DSA. Due to the exploratory nature of this study, no multiple comparison adjustments are performed.

## 3. Results

LTRs admitted to the hospital with moderate, severe, or critical COVID-19 gave consent after institutional review board approval for the proposed study between August 2020 and January 2021. All samples were collected and stored in the standard biorepository center in our institution. A total of 18 LTRs were included based on prior inclusion/exclusion criteria. None of the LTRs received vaccination before sample collection. All patients were treated for COVID-19 as per standard pre-established institutional guidelines for treating LTRs [7] (refer to supplement for COVID-19 treatment protocol).

### 3.1. Clinical Variables Related to COVID-19

Among the LTRs included, COVID-19 severity was classified as non-severe (*n* = 10) and severe (*n* = 8) as previously considered [12,13,14]. There were no major differences in baseline characteristics (Table 1 and Table 2). Most were Caucasian, with restrictive lung disease being the most common indication for lung transplantation. There were no significant differences in cumulative rejection score, presence of donor-specific antibodies, and chronic lung allograft dysfunction (CLAD) before COVID-19.

The median time interval between pre-COVID-19 and the first post-COVID-19 sample was 41.5 days. The median time between the COVID-19 positive test and the first post-COVID-19 sample was 1.5 days. There was also no significant difference in symptom duration before COVID-19 diagnosis in both groups. Due to a small sample size, the severe and critical COVID patients were grouped for comparative analysis of the outcomes, and the cohort was divided into non-severe (NS) and severe (S) groups, as previously considered [12,13,14].

In terms of COVID-19-specific outcomes, as shown in Table 3, compared with the NS group, the S group show higher levels of ferritin and D-dimer, although not significant. Acute respiratory failure was significantly noted in the S group (*p* < 0.001) with three deaths in the S group (Table 4, *p* = 0.06).

### 3.2. Auto-Antibodies and Characteristics

Serum from a total of 18 patients with pre-COVID-19 and post-COVID-19 samples were analyzed and the complete list of AAb changes was depicted (Figure 1). AAbs were clustered into three separate sub-groups representing interferon (IFN)-related, interleukin (IL)-related, and other groups for IgG classes of circulating AAbs (Table 5, Table 6 and Table 7; Appendix A). The longitudinal analysis also identified prominent and sustained increases in all 4 auto-antibodies against IFN at the second (PoC2) and third time-points (PoC3) (Appendix A).

### 3.3. IFN-Related AAbs

Levels of eight AAbs for IFN were significantly changed from pre to post-COVID-19 (Table 5). Six out of eight compared to pre-COVID-19 samples were elevated. AAbs were significantly associated with COVID-19 severity (*p* = 0.034). In linear regression analysis (Table 5), AAbs remained significantly associated with COVID-19 severity after controlling for time since transplant and the presence of DSA before COVID-19 (*p* = 0.02). 

### 3.4. IL-Related AAbs

AAbs for interleukins changed significantly in pre to post-COVID-19 analysis (*p* < 0.05) (Table 6). Among them, AAbs for interleukin 22 were significantly associated with COVID-19 severity (*p* = 0.002). Interestingly, in linear regression analysis, this remained significant after controlling for time since transplant and DSA presence before COVID-19. Also, striking reactivity of IL-8 and IL-17A auto-antibodies were exclusively noted in the deceased sub-category of S cohorts, which were sustained until the PoC3 Appendix A).

### 3.5. Other AAbs

Several AAbs such as those related to extracellular matrix proteins, TNF-related, centromere-related, and other miscellaneous AAbs changed significantly in the post-COVID-19 samples relative to pre-COVID-19 samples (Table 7) and remained sustained in the subsequent longitudinal PoC2 and PoC3 samples (Appendix A). However, none of them correlated significantly to COVID-19 severity. Additional linear regression analysis was carried out with all AAbs analyzed with COVID-19 severity revealed that a group of AAbs such as Collagen V, anti-mitochondrial antibodies (BCOADC-E2/OGDC-E2/PDC-E2), myositis specific antibodies (Mi-2) correlated significantly with COVID-19 severity after controlling for time since transplant and DSA presence before COVID-19 (Table 8). The list of AAbs that were significantly changed in association with either one of the variables such as disease severity, time since transplantation, or DSA was presented in the Appendix A.

## 4. Discussion

Pre-existing AAbs correlating with phenotypes of allograft injury, such as primary graft dysfunction [15] and Bronchiolitis Obliterans Syndrome (BOS) [16] are previously described by our group in LTRs. Although it is unclear about the mechanisms of AAbs and their pathogenic role, a better understanding is now underway and potential therapeutic strategies are being explored. In this context, we analyzed the presence of AAbs using a similar methodology and correlated with COVID-19 severity to identify potential biomarkers for disease severity. Additionally, we evaluated the COVID-19-associated role of AAbs in LTRs exclusively without the vaccine-related influences, since the samples used were all collected before any vaccines were made available to the public.

### 4.1. Interferon-Related AAbs Leading to Severe COVID-19 Disease

Viral infections break immune tolerance and cause autoimmunity by mechanisms such as molecular mimicry, bystander activation, and epitope spreading. Type I IFNs are essential for antiviral immunity [17,18,19]. In an international cohort, 3.5% of patients with severe COVID-19 carried rare inborn errors of type I IFN immunity [20]. In the same cohort, neutralizing AAbs were detected in 10% of patients with critical COVID-19 disease [5]. The authors suggest that the presence of these AAbs before COVID-19 suggests the possibility of them contributing to the severity. Additionally, multiple studies confirm the presence of AAbs to type I IFN in at least 10% of cases of critical COVID-19 disease in the general population [21,22,23,24,25,26,27]. Dysfunctional type I IFN immunity during early infection contributes to unrestrained viral replication and spread, resulting in uncontrolled hyperinflammation leading to life-threatening COVID-19 disease [28]. However, very few studies have described AAbs in LTRs who are immunosuppressed.

We describe AAbs to IFN-lambda that correlated significantly with COVID-19 severity in our cohort. IFN-lambda is critical for maintaining a balanced viral response in the respiratory tract. They are induced at lower viral burden and limit initial infection by inducing viral response.

Disruption of self-tolerance leading to autoimmunity and end-organ damage has been proposed to be the major driving force for severe manifestations of COVID-19 [29]. The persistence of such AAbs after the acute phase has been associated with “long COVID-19” [30,31]. Although there is concern for an increased risk of severe COVID-19 disease [32], the role of AAbs on COVID-19 severity in LTRs has not been described previously. Our study reports a substantial increase in several AAbs to IFN, interleukins, and a few others. In general, type I (IFN-alpha) and type III (IFN-lambda) are key innate antiviral factors that are quickly produced in response to an infection [33]. As in prior studies, delayed IFN response led to severe disease [34].

Transcriptional profiling of post-mortem lung samples from COVID-19-positive patients demonstrated no type 1 or type III IFN detected by RNA sequencing or semi-quantitative PCR [35,36]. Findings in our study demonstrate the presence of AAbs to both type 1 and type III IFN in a significant percentage of the deceased–severe group, suggesting a dysregulated IFN response leading to critical disease in LTRs. The presence of AAbs that neutralize IFN actions is one of the few mechanisms proposed in diminished or delayed IFN response [37] in those with critical disease. Andreakos et al. describe the potential therapeutic role of IFN-lambda in treating COVID-19 patients with severe disease [37]. Recent findings also describe that the genetic defects of the interferon type 1 system [19] and neutralizing antibodies against specific interferons [5] underlie critical COVID-19 disease, but in the non-transplant patient population.

Several AAbs are described in COVID-19 patients with no previous autoimmune disease (p-ANCA, c-ANA, Anti-RNP, etc.), but in a non-transplanted setting [17]. Autoimmune thrombocytopenia or anemia with COVID-19 disease [38] and or in some cases with vaccination, antiphospholipid syndrome, different forms of vasculitis, new systemic connective tissue disease such as systemic lupus erythematosus (SLE) are described as associated with COVID-19 disease [39]. In this realm, our study suggests that COVID-19 may not always be dealt with as a purely infectious disease but the presence of select AAbs suggestive of COVID-19-related autoimmunity despite immunosuppression makes the case for consideration of alternative therapies such as intravenous immunoglobulin, plasmapheresis for neutralization, or removal of pathogenic AAbs.

### 4.2. Interleukin-Related AAbs Leading to Severe COVID-19

In addition, among several cytokines that are described, IL-22 has a major impact on fibroblasts and epithelial cells. IL-22 is important for host protective immunity to both viral and bacterial infections [40]. IL-22 in COVID-19 is described to reduce the pneumonia severity via immune regulation and tissue protection [41,42]. Cagan et al. [43] describe mild COVID-19 disease in those with increased IL-22 expression. In our cohort of LTRs, we show that AAbs to IL-22 are correlated with severe COVID-19 disease. This implies that AAbs to IL-22 prevent the participation in epithelial protection or regeneration and prevent its role from being a crucial modulator for homeostasis and a regulator for host defense in the lung. The FDA has approved several research groups to study the efficacy of IL-22 during SARS-CoV-2 infection [40,44].

Thus, in the lung transplant cohorts, a lack of balance between pro-inflammatory and anti-inflammatory actions and loss of coordinated spatiotemporal cellular response of the immune system (associated with T- and B-cells) either individually or in combination could be thought of as a reason for the severe or critical form of COVID-19 disease. The above data strongly suggests that severe COVID-19 disease can be explained by AAbs. However, there is a need to understand the mechanisms as well as to determine whether these AAbs lead to severe disease or an epiphenomenon of marked inflammation [32].

### 4.3. Challenges with Lung Transplant Recipients

We demonstrated that AAbs are prevalent despite being on an immunosuppressive regimen and correlate with COVID-19 disease severity. LTRs with severe disease had greater than a two-fold increase in median ab-score, and this phenomenon was more pronounced in the severe–deceased group. Immunosuppression that includes calcineurin inhibitors, mTOR inhibitors, cell cycle inhibitors, and steroids in LTRs had a profound impact on mediators, such as cytokines, which result in infection and or acute rejection. A few cytokines’ genetic polymorphisms are reported to be associated with protection from infection [45]. It is still unclear why there is individual clinical variability in the course of COVID-19 disease in LTRs with similar immunosuppressive regimens ranging from silent infection to acute respiratory distress syndrome. Although there are clinical predictors such as body mass index, D-dimer, and pre-COVID-19 chronic lung allograft dysfunction (CLAD) for increased risk of death as described previously by our group [46], the role of AAbs described here implicate its underappreciated role in determining the course of COVID-19 in the LTR population and suggest its modulation to treat allograft function after lung transplantation. Post-viral immune response leading to a breach of self-tolerance leading to acute allograft injury is described in other community-acquired respiratory viral infections (CARV) in LTRs [47]. COVID-19 caused significant morbidity and mortality. More than 80% of cases reported needed hospitalization. Mortality ranged from 14 to 39% [1]. Several epidemiological associations and biological plausibility supported by animal models undoubtedly suggest several mechanisms by which immune response to CARV leads to acute or chronic lung injury. There is evidence that COVID-19 and other CARVs result in significant morbidity and mortality. Opportunities for future research exist in multiple areas, including early diagnosis and therapeutics, where there is a potential role for selective blockade of the pathways triggered by viral immune response resulting in tissue injury and allograft dysfunction.

### 4.4. Limitations of This Study

An apparent limitation of this study is that it is a retrospective study with a small sample size of LTRs from a single center without comparing healthy COVID-19 patients and or other immunosuppressed patients. This precludes generalizability and the determination of a clear and comprehensive pathogenic role played by AAbs in COVID-19 severity and its outcomes. In addition, due to the small sample size, the clustering of different COVID-19 severity categories together cannot be averted and thus, the influence of AAbs on gradation of severity could not be specifically established. Although the study has utilized microarray-based qualitative profiling of a total of 120 AAbs, given the breadth of autoantibody responses and their implications on different pathways associated with disease severity, prognosis, and outcomes, it merits high-throughput and unbiased proteome-scale investigations coupled with quantitative validations. Furthermore, it is not known whether all the AAbs detected have a functional role, or some represent an epiphenomenon.

## 5. Conclusions

In summary, we conclude that despite robust immunosuppression and expected T and B cell suppression, our data demonstrates the prevalence of pre-existing or de novo AAbs. Several AAbs are detected in LTRs with severe COVID-19. However, further studies are needed to determine whether these AAbs are involved in the pathogenesis of severe disease or an epiphenomenon of inflammation, since viral infection can trigger autoimmunity.

## Figures and Tables

**Figure 1 viruses-15-02045-f001:**
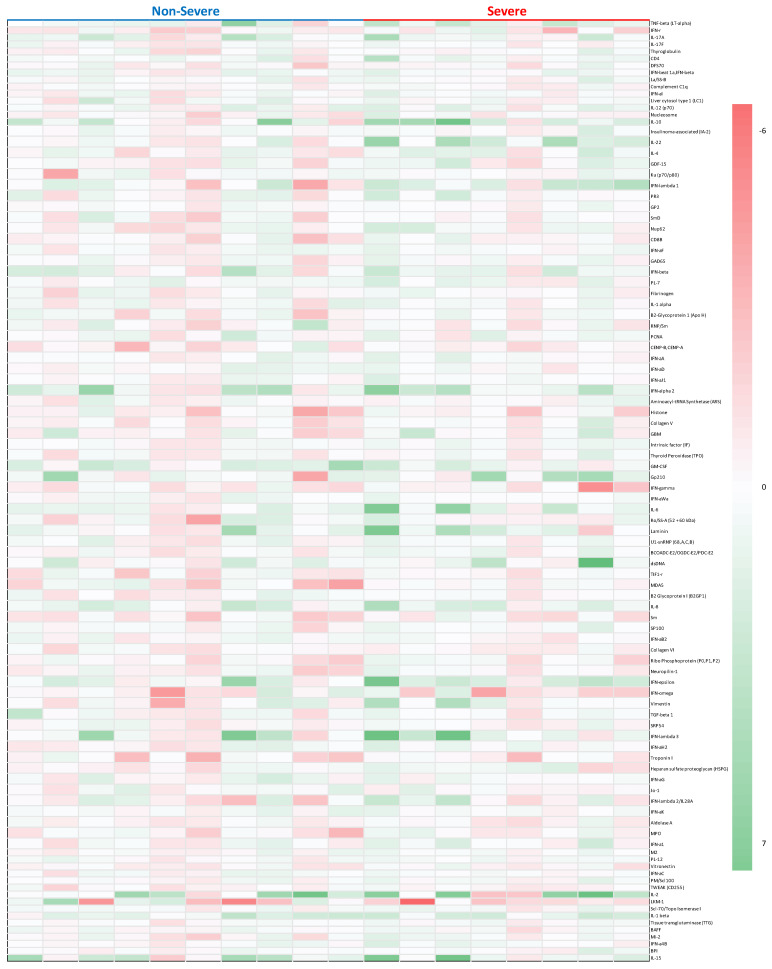
Antibody profile to viral antigens in LTRs with non-severe and severe COVID-19.

**Table 1 viruses-15-02045-t001:** Baseline Characteristics.

Variable	Non-Severe (N = 10)	Severe (N = 8)	All (N = 18)	*p*-Value
Age at Transplant				
mean ± sd	61 ± 9	61 ± 5		0.86
median (max, min) 95% CI	63 (42, 71) (54, 67)	60 (55, 72) (57, 66)		
BMI				
mean ± sd	28 ± 4	29 ± 6		0.73
median (max, min) 95% CI	28 (21, 36) (25, 31)	30 (17, 39) (24, 35)		
Gender				
Male	8 (80%)	6 (75%)	4 (22%)	0.80
Female	2 (20%)	2 (25%)	14 (78%)	
Race				
African American	0	2 (25%)	2 (11%)	1
Asian	1 (10%)	0	1 (6%)	
Caucasian	6 (60%)	5 (63%)	11 (61%)	
Hispanic	3 (30%)	1 (13%)	4 (22%)	
Transplant Indication				
Obstructive	3 (30%)	0	3 (17%)	0.21
Restrictive	7 (70%)	8 (100%)	15 (83%)	
Type of Transplant				
Double	8 (80%)	5 (63%)	13 (72%)	0.61
Single	2 (20%)	3 (38%)	5 (28%)	
Time Since Transplant (days)				
mean ± sd	1557 ± 1287	1388 ± 987		0.77
median (max, min) 95% CI	1145 (115, 4182) (636, 2478)	1110 (229, 2979) (563, 2213)		
Hypertension (Y)	9 (90%)	8 (100%)	17 (94%)	1
DM (Y)	6 (60%)	4 (50%)	10 (56%)	1
Hyperlipidemia (Y)	8 (80%)	7 (87%)	15 (83%)	1
OSA (Y)	2 (20%)	2 (25%)	4 (22%)	1
CAD (Y)	4 (40%)	4 (50%)	8 (44%)	1
CHF (Y)	0	1 (12%)	1 (6%)	0.44

Y—Yes; Categorical variables: Fisher’s Exact; continuous variables: *t*-test. DM—Diabetes Mellitus; OSA—Obstructive Sleep apnea; CAD—Coronary artery disease; CHF—Congenital heart disease. The minimum and maximum range between the COVID-19 positive test and the first post-COVID-19 sample was zero and 8 days with the first quartile falling on day 0 and the third quartile falling on the third day.

**Table 2 viruses-15-02045-t002:** Treatment effects in the severe and non-severe COVID patients.

Variable	Non-Severe (N = 10)	Severe (N = 8)	All (N = 18)	*p*-Value
Immunosuppression				
Prograf, Cyclosporin, Sirolimus	10 (100%)	7 (100%)	17 (100%)	1.00
Prograf (missing = 1)	10	7	17	
Immunosuppression				
Imuran, Cellcept, Myfortic	10 (100.00%)	8 (100.00%)	18 (100.00%)	1.000
Azathioprine	2	1	3	
Cellcept	8	5	13	
Myfortic	0	2	2	
Prednisone dose at COVID DX (mg)				
mean ± sd	7.500 ± 3.118	8.750 ± 3.273		0.4205
median (max, min) 95% CI	7.50 (5, 15) (5.270, 9.731)	8.75 (5, 15) (6.014, 11.487)		
History of CMV PCR+	1 (10.00%)	4 (50.00%)	5 (27.78%)	0.1176
History of EBV PCR+	3 (30.00%)	4 (50.00%)	7 (38.89%)	0.6305
Cumulative Acute Rejection Score				
mean ± sd	1.500 ± 1.780	0.250 ± 0.463		0.0575
median (max, min) 95% CI	1 (0, 6) (0.227, 2.773)	0 (0, 1) (-0.137,0.637)		
History of DSA Positive Pre-COVID	5 (50.00%)	3 (37.50%)	8 (44.44%)	0.6641
DSA MFI				
mean ± sd	3622.4 ± 4692.2	4767.0 ± 442.4		0.6162
median (max, min) 95% CI	1600 (1143, 11,978) (−2203.7, 9448.5)	4720 (4350, 5231) (3368.1, 5865.9)		
DSA at COVID	3 (30.00%)	0	3 (16.67%)	0.2157
Pre-COVID CARV	8 (80.00%)	4 (50.00%)	12 (66.67%)	0.3213
Pre-COVID CLAD	3 (30.00%)	3 (37.50%)	6 (33.33%)	1.0000
CLAD TYPE				
NA	6 (60.00%)	4 (50.00%)	10 (55.56%)	1.0000
Obstructive	2 (20.00%)	2 (25.00%)	4 (22.22%)	
Restrictive	2 (20.00%)	2 (25.00%)	4 (22.22%)	
Symptom duration before COVID DX				
mean ± sd	2.200 ± 1.033	3.250 ± 2.376		0.2735
median (max, min) 95% CI	2 (1, 4) (1.4312, 2.9388)	3 (0, 7) (1.264, 5.236)		

Categorical variables: Fisher’s Exact; continuous variables: *t*-test. DSA—Donor-specific antibodies; CMV—Cytomegalovirus; EBV—Ebstein Barr Virus; MFI—Mean Fluorescence Intensity; CARV—Community-Acquired Respiratory Viral Infections; CLAD—Chronic Lung Allograft Dysfunction; DX—Diagnosis; *p*-value < 0.05 is considered statistically significant.

**Table 3 viruses-15-02045-t003:** COVID-19-specific outcomes.

Variable	NS (N = 10)	S (N = 8)	All (N = 18)	*p*-Value
Ferritin				
mean ± sd	375.0 ± 416.9	1337.0 ± 1036.5		0.0361
median (max, min) 95% CI	257 (46, 1262) (76.8, 673.2)	1287 (201, 3083) (470.5, 2203.5)		
D-Dimer				
mean ± sd	0.566 ± 0.410	1.145 ± 0.883		0.1195
median (max, min) 95% CI	0.410 (0.250, 1.510) (0.273, 0.859)	0.710 (0.310, 2.880) (0.406, 1.884)		
ALC				
mean ± sd	0.548 ± 0.346	0.520 ± 0.388		0.8736
median (max, min) 95% CI	0.405 (0.300, 1.450) (0.300, 0.756)	0.435 (0.190, 1.390) (0.196, 0.844)		
Co-morbid kidney dysfunction (Y)	4 (40.00%)	5 (62.50%)	9 (50.00%)	0.6372
Spirometry Decline > 10% at COVID (Y)	3 (42.86%)	1 (33.33%)	4 (40.00%)	NA

NS—non-severe COVID-19 (mild to moderate) (total N = 10); S—severe COVID-19 (severe to critical) (total N = 8); Y—Yes; Categorical variables: Ferritin (ng/mL), D-dimer (mg/L FEU) and ALC—Absolute lymphocyte count (×10^9^/L); Fisher’s Exact; continuous variables: *t*-test; NA—*p*-value is not appropriate. *p*-value < 0.05 is considered statistically significant.

**Table 4 viruses-15-02045-t004:** COVID-19-specific functional outcomes.

Variable	NS (N = 10)	S (N = 8)	All (N = 18)	*p*-Value
Acute Respiratory Failure (Y)	0	8 (100.00%)	8 (44.44%)	<0.0001
Hospital Length of Stay				
mean ± sd	7.000 ± 2.582	9.714 ± 4.923		0.1570
median (max, min) 95% CI	6.5 (4, 11) (5.153, 8.847)	10.0 (2, 18) (5.161, 14.268)		
Readmission within 30 Days (Y)	3 (30.00%)	3 (37.50%)	6 (33.33%)	1.0000
Death (Y)	0	3 (37.50%)	3 (16.67%)	0.0686

Y—Yes; Categorical variables: Fisher’s Exact. *p*-value < 0.01 and 0.0001 are considered statistically significant.

**Table 5 viruses-15-02045-t005:** Changes in interferon-related AAbs in LTRs before and after COVID-19 and correlation to COVID-19 severity.

	All Patients (N = 18)			NS (N = 10)		S (N = 8)		*p* < 0.05
	Mean	SD	*p*-Value	Mean	SD	Mean	SD	*p*-Value
IFN-alpha2 (Pre–Post)	1.686	1.955	0.002	1.332	2.089	2.13	1.807	0.406
IFN-beta-1a (Pre–Post)	0.399	0.687	0.025	0.352	0.782	0.458	0.595	0.755
IFN-beta (Pre–Post)	0.806	1.384	0.024	0.693	1.599	0.947	1.15	0.712
IFN-epsilon (Pre–Post)	1.52	1.898	0.003	0.891	1.66	2.305	1.984	0.119
IFN-gamma (Pre–Post)	−0.804	1.294	0.017	−0.609	0.827	−1.048	1.75	0.528
IFN-lambda 1 (Pre–Post)	0.633	1.907	0.177	−0.197	1.843	1.671	1.499	0.034
IFN-lambda 3 (Pre–Post)	1.667	2.738	0.019	1.147	2.537	2.317	3.01	0.384
IFNR (Pre–Post)	−0.639	1.168	0.033	−0.704	0.917	-0.557	1.49	0.8

NS—Non-severe COVID-19; S—Severe COVID-19; AAbs—Autoantibody; SD—Standard Deviation; Pre—Pre-COVID-19; Post—Post-COVID-19. IFN—interferon. *p*-value < 0.05 is considered statistically significant.

**Table 6 viruses-15-02045-t006:** Changes in interleukin-related AAbs in LTRs before and after COVID-19 and correlation to COVID-19 severity.

	All Patients (N = 18)			NS (N = 10)		S (N = 8)		*p* < 0.05
AAb	Mean	SD	*p*-Value	Mean	SD	Mean	SD	*p*-Value
IL-1 beta (Pre–Post)	1.384	1.025	0	1.269	1.077	1.527	1.01	0.611
IL-2 (Pre–Post)	2.303	3.14	0.006	1.958	2.64	2.734	3.822	0.617
IL-6 (Pre–Post)	1.176	1.811	0.014	0.52	0.844	1.996	2.38	0.131
IL-8 (Pre–Post)	1.086	1.177	0.001	0.903	1.099	1.315	1.306	0.477
IL-10 (Pre–Post)	1.657	2.441	0.01	0.911	2.267	2.588	2.465	0.153
IL-15 (Pre–Post)	1.741	2.396	0.007	1.563	2.111	1.962	2.847	0.737
IL-17A (Pre–Post)	0.911	1.461	0.017	0.861	1.491	0.973	1.521	0.877
IL-22 (Pre–Post)	0.993	1.8	0.032	-0.081	0.932	2.335	1.744	0.002

NS—Non-severe COVID-19; S—Severe COVID-19; AAbs—Autoantibody; SD—Standard Deviation; Pre—Pre-COVID-19; Post—Post-COVID-19. IL—interleukin. *p*-value < 0.01 is considered statistically significant.

**Table 7 viruses-15-02045-t007:** Matrix proteins, growth factors status before and after COVID-19, and correlation to COVID-19 severity.

		All Patients (N = 18)			NS (N = 10)		S (N = 8)		*p* < 0.05
Category	AAb	Mean	SD	*p*-Value	Mean	SD	Mean	SD	*p*-Value
**Extracellular matrix protein**	Collagen-VI (Pre–Post)	−0.375	0.61	0.018	−0.349	0.725	−0.407	0.475	0.847
	Laminin (Pre–Post)	1.041	2.016	0.043	0.6	1.549	1.592	2.482	0.314
	Vitronectin (Pre–Post)	−0.332	0.628	0.039	−0.463	0.564	−0.167	0.704	0.336
**TNF-** **related**	TNF-β (LT-alpha) (Pre–Post)	1.168	1.588	0.006	0.887	1.831	1.519	1.249	0.418
	TWEAK (CD255) (Pre–Post)	−0.319	0.596	0.036	−0.367	0.701	−0.259	0.474	0.716
**Centromere** **related**	CENP-B, CENP-A (Pre–Post)	−0.611	0.954	0.015	−0.734	1.196	−0.458	0.569	0.557
	Histone (Pre–Post)	−0.747	1.301	0.026	−0.91	1.476	−0.544	1.108	0.57
**Other** **miscellaneous**	CD-4 (Pre–Post)	0.533	0.953	0.03	0.325	0.723	0.794	1.181	0.315
	GM-CSF (Pre–Post)	1.11	1.29	0.002	1.406	1.425	0.74	1.072	0.29
	Sm (Pre–Post)	−0.714	1.01	0.008	−0.9	1.047	−0.481	0.978	0.399
	Troponin I (Pre–Post)	−0.737	1.248	0.023	−0.903	1.444	−0.531	1.009	0.546

NS—Non-severe COVID-19; S—Severe COVID-19; AAbs—Autoantibody; SD—Standard Deviation; Pre—Pre-COVID-19; Post—Post-COVID-19. TNF—Tumor necrosis factor; TWEAK—cell surface-associated type II transmembrane protein; CENP-B, CENP-A—centromere-related protein; CD-4—cluster of differentiation 4; GM-CSF—Granulocyte-macrophage colony-stimulating factor; Sm—spliceosomal protein.

**Table 8 viruses-15-02045-t008:** Univariate linear regression model significantly altered AAbs in association with disease severity in LTRs with COVID-19 disease.

Antibody	Variable	Estimate ± SE	96% CI	*p*-Value
BCOADC-E2/OGDC-E2/ PDC-E2	COVID-19 Severity	0.7112 ± 0.2959	(0.0766, 1.3458)	0.0307
	Time Post-Transplant	0.0004 ± 0.0001	(0.0001, 0.0006)	0.0173
	History of DSA Prior to COVID	−0.0490 ± 0.2956	(−0.6830, 0.5851)	0.8708
Collagen V	COVID-19 Severity	0.8743 ± 0.3571	(0.1085, 1.6401)	0.0281
	Time Post-Transplant	0.0001 ± 0.0002	(−0.0002, 0.0005)	0.4296
	History of DSA Prior to COVID	0.7489 ± 0.3567	(−0.0162, 1.5139)	0.0544
IFN-lambda 1	COVID Severe	2.0666 ± 0.7962	(0.3589, 3.7742)	0.0212
	Time Post-Transplant	0.0003 ± 0.0004	(−0.0005, 0.0010)	0.4931
	History of DSA Prior to COVID	1.2483 ± 0.7954	(−0.4577, 2.9544)	0.1389
IL-22	COVID Severe	2.5713 ± 0.6348	(1.2098, 3.9328)	0.0012
	Time Post-Transplant	0.0003 ± 0.0003	(−0.0003, 0.0009)	0.3656
	History of DSA Prior to COVID	0.8818 ± 0.6342	(−0.4784, 2.2420)	0.1861
Mi-2	COVID Severe	0.8121 ± 0.3819	(−0.0070, 1.6311)	0.0517
	Time Post-Transplant	0.0004 ± 0.0002	(0.0000, 0.0007)	0.0576
	History of DSA Prior to COVID	0.3298 ± 0.3815	(−0.4885, 1.1480)	0.4020

*p*-value < 0.05, 0.01 are considered statistically significant. IFN—interferon; IL—interleukin; BCOADC-E2—Branched chain 2-oxo acid dehydrogenase complex; OGDC-E2—2-oxoglutarate dehydrogenase complex; PDC-E2—mitochondrial pyruvate dehydrogenase complex; Mi-2—dermatomyositis-specific autoantigen.

## Data Availability

Due to ethical reasons, the data is not publicly available. However, upon request, the data presented in this study can be shared.

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
