# Peer review of "Crossroads between Autoimmunity and COVID-19 in Lung Transplant Recipients"

_viruses, 2023, doi:10.3390/v15102045_

Round 1
Reviewer 1 Report
In this manuscript, Narasimhan et al. have performed a retrospective study on lung transplant recipients (LTR) with COVID-19 and analyzed several autoantibodies (AAbs) before and after infection. The authors noted that autoantibodies related to interferon lambda (IFN λ) and IL22 are significantly increased from pre-COVID-19 to post-COVID-19, and they are associated with the severity of infection in LTR. Overall, the study is important given the low availability of lung transplant recipients and COVID complications however, to generalize the phenomenon of AAbs in the severity of COVID-19 in LTR, a bigger cohort is needed to study. I have the following comments.
1-The authors have shown a significant increase in the AAbs to IFN λ and IL22 and found an association of increased severity of COVID-19 outcome in LTR. Whether the authors have checked the levels of these cytokines in the patient? Whether the level of these cytokines, in particular, is reduced in the more severe COVID-19 group.
2- It is unclear from the text how the lung transplant was affected in patients with higher levels of autoantibodies.
3- It is also not clear whether the cause of death in LTRs with higher AAbs resulted from COVID-19-related complications or was due to failure of the transplant/transplant rejection.
4- For paragraphs 3.3 and 3.4, the author should use AAbs instead of IL-22 or IFN λ alone in the text, as it is confusing whether the authors are referring to AAbs or the cytokine itself.
Reviewer 2 Report
1. These authors have measured the development of autoantibodies in patients with lung transplants who subsequently had COVID-19 infection. They were trying to determine if there was an association between the development of an autoantibody against host defense factors, such as interferon and interleukins, and the development of severe COVID-19 infection. Clearly, this is a relatively unique data set.
2. I think the authors assume that the reader has a significant background in transplantation and immunology. Some of the ideas in this paper need to be clarified. First, what are donor specific antibodies and what is their importance? Second, is the cumulative acute rejection score a digit, such as 0, 1, 2 etc. Third, how is chronic lung allograft dysfunction defined?
3. Most of the tables have do not have an adequate number of footnotes to explain the abbreviations used in the tables.
4. They reported the median that time for the first post COVID sample after a positive PCR of 1.5 days. They need to add information regarding quartile 1, quartile 3, and the maximum-minimum range.
I am assuming that the authors do not think that the COVID infection change the level of autoantibodies. Is that a safe assumption?
5. They should probably include the definition severe COVID infection in the methods or the results.
6. Is there a level of autoantibody which is known to have clinical significance either in experimental or clinical studies?
7. In the tables they noted that red bold numbers indicate a peak value less than 0.05. In fact, some of the values are above 0.05.
8. The authors might explain the basis for their linear regression models. It is not clear to me as to the utility of controlling for time from transplant and a history of DSA
9. The statistical comparisons involve the use of T tests for small groups of patients who likely do not have a normal distribution of the analyzed variables. Is this an adequate approach?
Double check noun verb agreement
Reviewer 3 Report
In this article, Narashiman et al. investigated the changes of levels of autoantibodies in lung transplant recipients before and after COVID-19. This is an interesting topic, however there some comments I would like to share with the authors:
1. first occurrence of CAR is misspelled,
2. in line 88 there is a reference to a protocol which is not included in the supplementary material,
3. too many digits of numbers in table (e.g., age and BMI one digit is enough, percentage of patients – no digit after decimal point is necessary),
4. some not common abbreviations are defined at first appearance (e.g., OSA, CAD, CHF, ALC),
5. table 1a and 1b have the same title,
6. table 3b and 3c have the same title,
7. red bold lettered indication of p values<0.05 in tables is not consistent (some values >0.05 are red, some values <0.05 are not red),
8. in table 2a ferritin, D-dimer, ALC have no units,
9. Figure A1: too many colors, using shades of green and red for decreased and increased antibody levels would be more helpful; it is not clear whether differences of signal-to-noise ratio or normalized net fluorescence intensity were used,
10. no multiple comparison adjustments were used: there can be several false positive results considering the 120 antibodies and other factors analyzed,
11. in line 159 D-dimer level is stated to be higher in the severe COVID-19 group, however the difference is not significant, as shown in Table 2a (p=0.1195).
12. there is no interpretation of results shown in Supplementary Table A1-3,
13. it is not clear whether differences of signal-to-noise ratio or normalized net fluorescence intensity were used to compare antibody levels,
14. it is not clear why and how AAbs were clustered into three separate sub-groups,
15. univariate linear regression was used to control time since transplant and presence of DSA prior to COVID-19, but this is not appropriate for this purpose (should be multivariate analysis). Univariate linear regression analysis gives compatible results with paired t-test,
16. it is not clear how not severe (NS) and severe (S) groups were compared (only two columns of p values in Tables 3a-c). Before versus after in each group or NS versus S?
